# Novel Insights into the Initiation, Evolution, and Progression of Multiple Myeloma by Multi-Omics Investigation

**DOI:** 10.3390/cancers16030498

**Published:** 2024-01-24

**Authors:** Lixin Gong, Lugui Qiu, Mu Hao

**Affiliations:** 1State Key Laboratory of Experimental Hematology, National Clinical Research Center for Blood Diseases, Haihe Laboratory of Cell Ecosystem, Institute of Hematology & Blood Diseases Hospital, Chinese Academy of Medical Sciences & Peking Union Medical College, No. 288 Nanjing Road, Tianjin 300020, China; gonglixin@ihcams.ac.cn; 2Tianjin Institutes of Health Science, Tianjin 300020, China; 3Gobroad Healthcare Group, Beijing 100072, China

**Keywords:** multiple myeloma, single-cell sequencing, tumor initiation, tumor evolution

## Abstract

**Simple Summary:**

The integration of genomics, transcriptomics, epigenomics and proteomics broad our understanding of the disease. In this review, we generally introduce the progress in single-cell multi-omics technologies. We summarize the novel insights into the initiation, evolution, and progression of multiple myeloma when using diverse multi-omics sequencing tools. We intend to discuss the challenges and future perspectives of omics in myeloma. Based on these recent omics findings in myeloma, the advanced single-cell multi-omics technologies allow the researchers to acquire both the integral overview of the tumor environment and also the sophisticated internal contexture of the tumor landscape.

**Abstract:**

The evolutionary history of multiple myeloma (MM) includes malignant transformation, followed by progression to pre-malignant stages and overt malignancy, ultimately leading to more aggressive and resistant forms. Over the past decade, large effort has been made to identify the potential therapeutic targets in MM. However, MM remains largely incurable. Most patients experience multiple relapses and inevitably become refractory to treatment. Tumor-initiating cell populations are the postulated population, leading to the recurrent relapses in many hematological malignancies. Clonal evolution of tumor cells in MM has been identified along with the disease progression. As a consequence of different responses to the treatment of heterogeneous MM cell clones, the more aggressive populations survive and evolve. In addition, the tumor microenvironment is a complex ecosystem which plays multifaceted roles in supporting tumor cell evolution. Emerging multi-omics research at single-cell resolution permits an integrative and comprehensive profiling of the tumor cells and microenvironment, deepening the understanding of biological features of MM. In this review, we intend to discuss the novel insights into tumor cell initiation, clonal evolution, drug resistance, and tumor microenvironment in MM, as revealed by emerging multi-omics investigations. These data suggest a promising strategy to unravel the pivotal mechanisms of MM progression and enable the improvement in treatment, both holistically and precisely.

## 1. Introduction

The occurrence of MM is a multi-step process evolving from premalignant clinical phases, monoclonal gammopathy of uncertain significance (MGUS), and smoldering multiple myeloma (SMM), to symptomatic multiple myeloma. The epidemiological statistics show that MGUS affects an estimated 3% to 5% of people aged over 65 years old and transforms to MM at a rate of 1% to 2% per annum [1]. At the SMM stage, patients are estimated to have an average risk of progression of 10% per year for the first 5 years. The cumulative probability of progression will reach 73% at 15 years [2]. Therefore, lifelong monitoring of SMM is recommended by most clinical guidelines [3]. Several clinical trials also indicate the benefit in outcomes translated by initiation of therapy at the SMM stage compared with watchful monitoring. The QuiRedex study (ClinicalTrials.gov ID: NCT00480363) showed that compared to observation, early treatment with lenalidomide plus dexamethasone significantly delayed progression to overt MM, and also had better 3-year overall survival in patients with high-risk SMM [4,5,6]. Lonial et al. also demonstrated that early intervention with lenalidomide monotherapy could also delay progression to symptomatic MM compared with observation in patients with intermediate- or high-risk SMM [7]. The iStopMM study (ClinicalTrials.gov ID: NCT03327597) investigating the potential benefits and harms of screening for precursor stages of MM in Iceland is ongoing [8]. It has yet to be seen whether pioneering intervention at MGUS stage will further benefit patients. Therefore, the intervention of premalignant stages is at the forefront of disease monitoring. However, it remains a challenge to predict the progression of MGUS/SMM to overt MM.

The concept of cancer stem cells (CSCs) or the tumor-initiating cells (TICs) has been proposed for four decades. At first, CSCs were formally tested in acute myeloid leukemia and stated as the CD34^+^/CD38^−^ subpopulation. Then the CSC population was identified in diverse types of hematological and solid malignancies, such as breast cancer, glioma, liver cancer, pancreas cancer, and colorectal cancer, though CSC marker expression is not uniform across different tumor types. The CSC model highlights that a minor population of tumor cells hidden in cancers are more aggressive malignant cells which cause and fueling tumor growth. The existence of CSCs explains the inevitable recurrence of tumors, which is analogous to clinical observations in MM patients. However, whether CSCs exists in the context of MM is debatable. The CSC theory has inspired the investigations into exterminating the minor residual tumor cells, the probable origin of therapy-resisting and disease recurrent clones in MM [9,10].

Malignant transformation of a normal cell is a multistep and multifaceted process. Clonal evolution drives MM progression, enabling tumor heterogeneity based on selective internal and external pressures [11]. Genetic lesions, epigenetic modification, and tumor microenvironment interact together to favor a competitive advantage for transformed cells and promote the tumor development initiative. Novel technological developments allowing for the genetic information at single-cell resolution are opening new realms of discovery (Figure 1). Multi-omics single-cell characterization and integrated sequencing information of tumor cells provide a complex holistic picture of ongoing tumor dynamics and unravel the molecular mechanisms underlying tumor initiation and evolution. Here, we review the exploration into tumor-initiating cells in MM and the recent advances in the understanding of tumor evolution by multi-omics single-cell profiling.

## 2. Single-Cell Sequencing Technologies

### 2.1. Introduction to the Progress in Single-Cell Sequencing Technologies

Conventional bulk population sequencing can provide only the average signal for an ensemble of cells. Rapid progress in single-cell sequencing technologies and analytical methodologies has provided a plethora of valuable insights into complex biological systems. Generally, the single-cell analyses allow researchers to uncover novel biological discoveries relative to traditional methods that assess bulk cell populations: 1. The ability to detect the rare cell populations or specific cellular states has potential implications for furthering our understanding the exquisite biological systems. 2. The advanced computational methodologies enables the researchers to deconvolute highly diverse immune cell compositions in heathy and disease status. 3. Cell lineage can be determined by bioinformatics pipelines, such as delineating lymphocyte fate or defining cell differentiation route [12]. The first single-cell transcriptome sequencing based on a next-generation sequencing platform was published in 2009 by Tang et.al. [13]. Since this study, there entered an era of rapidly progressed sequencing methodology at high-resolution and an explosion of views in single-cell heterogeneity on a global scale. 

### 2.2. Single-Cell Genomics

Single-cell DNA sequencing (scDNS-seq) technology has been developed to identify the individual mutations, mainly including single nucleotide variants (SNVs) and copy number aberrations (CNAs) that occur in one cell. Due to the limited quantity of a single cell, whole-genome amplification (WGA) technologies are needed for sequencing an entire genome of a cell. The three major methods of WGA include degenerate oligonucleotide-primed polymerase chain reaction (DOP-PCR), multiple displacement amplification (MDA), and multiple annealing and looping-based amplification cycles (MALBAC) [14]. However, the amplification biases decrease coverage uniformity and further obscure the detection of CNAs. Furthermore, the depth of sequencing coverage and the throughput are also important specifications of scDNA-seq [15,16].

Single-cell RNA sequencing (scRNA-seq) has been widely applied to a variety of studies. The common procedures of scRNA-seq include single-cell capture, reverse transcription, cDNA synthesis, and sequencing. Nowadays, a series of sequencing technologies have been established, such as STRT-seq, SMART-seq, MARS-seq, Drop-seq, CEL-seq, and 10× Genomics, etc. Here, we mainly introduce two frequently used platforms, namely SMART-seq and 10× Genomics sequencing. The 10× Genomics Chromium is based on a microfluidics approach with high throughput. It can capture from 5000 to 10,000 single cells and prepare a cDNA library. However, it concentrates exclusively on the 5′ or 3′ ends of the transcript with a bias [17]. SMART-seq is a full-length sequencing approach and can detect mRNA isoforms, though the throughput is relatively lower than the 10× Genomics platform. [18]. Single-nucleus RNA sequencing (snRNA-seq) was firstly introduced by Grindberg et al. in 2013, and eliminates the potential transcriptomic changes during the single-cell isolation procedure [19]. SnRNA-seq further increases the availability of sequencing samples and makes the sample preparation simpler. SnRNA-seq can also provide more in-depth gene information, including intron region and inter-gene region sequencing data. However, it should be noted that the capture of RNA only the from nucleus will definitely ignore some important biological information [20]. Therefore, when faced with multiple sequencing methods with different advantages, the choice of the sequencing platform should be oriented by experimental objectives.

### 2.3. Single-Cell Epigenomics

Epigenetic regulation explains how the same sequence of DNA endows varying expression patterns in different cells. Epigenetic features include DNA methylation, chromatin accessibility, histone modifications, chromatin interactions, and chromatin 3D architecture. Single-cell epigenomic techniques can overcome cellular heterogeneity and unravel cell type-specific differences and dynamics. However, the current single-cell epigenomic assays face some problems, such as low throughput, limited coverage per cell, elevated costs, amplification bias, DNA damage upon sample processing, and differences in library sizes across samples [21]. 5-Methylcytosin (5mC) is the predominant form of DNA methylation mostly in the context of CpG dinucleotides. DNA methylation plays a critical role in restricting gene expression [22]. In particular, dysregulated methylation of tumor suppressors or oncogenes have crucial implications in cancer. The first bisulfite-based single-cell sequencing technologies, single cell-reduced representation bisulfate sequencing (scRRBS), was launched in 2013 [23,24]. ScRRBS is a cost-effective approach that offers high coverage profiling of CpG-rich regions. Furthermore, multiple bisulfite-based single-cell sequencing technologies have been developed, facilitating the increase in CpG coverage. The whole genome approaches, such as scWGBS, further enable the characterization of DNA methylation states at both promoters and promoter-distal cis-regulatory elements (CREs) and can detect up to 50% of CpG dinucleotides per cell. However, scWGBS has a relatively high sequencing cost and low throughput [25]. Additionally, methylation sequencing approaches based on methylation-sensitive restriction enzymes (MSREs) have been developed. These approaches can ensure better DNA integrity in the absence of bisulfite, such as single-cell CpG-island sequencing (scCGI-seq), epigenomics and genomics of single cells analyzed by restriction (epi-gSCAR) and single cell-targeted analysis of the methylome (scTAM-seq) [26]. Chromatin accessibility reflects both transcription factors binding and the regulatory potential of a genetic locus. Single-cell assay for transposase-accessible chromatin (scATAC-seq) is the most used technique to investigate chromatin accessibility, which uses hyperactive Tn5 transposase to insert sequencing adaptors into accessible chromatin regions [27]. Nowadays, in combination with scRNA-seq, the combinatorial indexing platform or droplet-based microfluidic platform have further offered scATAC-seq exceptional throughput and sequencing library complexity, such as sciATAC-seq, scATAC-seq by 10× Genomics, and Bio-Rad Laboratories [28,29]. These techniques will contribute to revealing CREs that are accessible in rare cell types and will unravel the dynamics of CREs during development of disease. Post-translational modifications in histones can lead to transcriptionally permissive and repressive chromatin states. ChIP-seq has been a widely used method for profiling histone modification and transcription factor binding in bulk samples. Approaches for detecting histone modifications and DNA-protein interactions at single-cell resolution have emerged, including scChIP-seq, single-cell chromatin immunocleavage sequencing (scChIC-seq), and single-cell cleavage under targets and tagmentation (scCUT&Tag). The intricate 3D organization of the genome is essential for normal cellular functions, from gene expression to DNA replication. For example, chromatin 3D architecture enables distal enhancers to be positioned close to their target genes and regulates gene expression [30,31]. Currently, studies in 3D genomes are dependent on two main technologies: imaging, that is, fluorescence in situ hybridization of DNA (DNA-FISH), and methods based on chromosome conformation capture (3C) [32]. Although DNA-FISH has advantages in visualizing the spatial organization of chromosomes and genes in the nucleus at single-cell resolution, it is a low-throughput method. High-throughput chromosome conformation capture (Hi-C) utilizes proximity ligation of DNA regions and allows genome-wide analysis of chromatin contacts. Single-cell Hi-C (scHiC) has also been developed [21].

### 2.4. Single-Cell Proteomics

While genomes and transcriptomes can be delineated at the single-cell level, single-cell profiling of proteomes is not yet well established. As the post-translational modifications, alternative splicing and germline variants can result in a myriad of proteoforms, it is a big challenge to analyze the complex protein mixtures. The emerging landscape of single-molecule protein sequencing and fingerprinting technologies is already vast. Mass spectrometry (MS) remains the main approach for protein identification and continues to develop toward single-cell resolution. Also notably, metabolite detection also highly relies on the MS technology. The improvement in MS technology will also benefit the metabolomics research [33]. The major limitation of progress in single-cell MS (scMS) is that proteins cannot be amplified by a process analogous to PCR for oligonucleotides. Recently the success of single-cell proteome analysis is based on the high protein content of large cells, such as human oocytes and Xenopus embryos. Thus, the key areas of development for scMS include the minimization of the loss of proteins or peptides during sample preparation and increase the current limited throughput for the number of cells to be analyzed [34]. How to increase throughput of scMS is another challenge. The current approach to solve this problem is to use a multiplexed quantitative scMS approach. Each single-cell proteome is labeled with a unique mass barcode similar to the addition of a DNA barcode for NGS methods, so the multiple samples can be combined and analyzed simultaneously. However, the current availability of mass barcodes for scMS only enables the analysis of up to 18 samples due to the difficulty in finding compatible chemical tags. Single Cell ProtEomics (SCoPE2) using an isobaric carrier enables the peptides from a single cell to be labeled by tandem mass tags (TMTs or TMTpro) for multiplexed analysis. SCoPE2 allows analyzing ~200 single cells per 24 h, further increasing the throughput and the proteomic depth [35,36]. Additionally, multimodal measurements of single cells are expected to provide vast information for normal biological processes and disease treatment. For example, single-cell proteomics utilizing laser capture microdissection is used to produce spatial proteome maps of specific tissue. CITE-seq is now one popular method combining single-cell antibody-based proteomics with RNA transcript detection in microfluidic droplets [37]. In recent years, genomic variations and chromatin profiles at single-cell resolution are also shown to be compatible with antibody-based protein detection, such as DAb-seq, ASAP-seq, ICICLE-seq (integrated cellular indexing of chromatin landscape and epitopes), NEAT-seq (single-cell sequencing of nuclear protein epitope abundance, chromatin accessibility, and the transcriptome), and TEA-seq (transcriptomes, epitopes, and ATAC by sequencing). Importantly, a method called nanodroplet splitting for linked multimodal investigations of trace samples (nanoSPILTS) has recently been reported. NanoSPLITS achieved parallel measurement of transcriptomes and proteomes from the same single cells by dividing a cell-containing droplet for scMS and scRNA-seq analysis simultaneously. Therefore, as scMS approaches maturity, a multimodal measurement assay in single cells will definitely improve our understanding of steady state physiology and therapeutic interventions [38]. 

## 3. Tumor Initiation

### 3.1. Phenotypic Characteristics of Tumor-Initiating Cells in MM

The high occurrence of relapse in MM has led to the speculation that the existence of pathogenic myeloma initiating/progenitor cell populations result in the recurrence of the disease. Nevertheless, a comprehensive understanding of the surface markers, phenotypic characteristics, and cellular biology associated with these MM cell populations remains elusive. The pertinent studies on the origin of MM have been summarized in Table 1.

In the B cell differentiation process, the germinal center (GC) B cell can differentiate into either a memory B cell or a plasma cell. A plasma cell (PC) is a terminally differentiated cell type in the B cell lineage. Different stages of B cell development are accompanied by multiple changes in immunophenotype and regulators of differentiation [39]. Patients harboring less-differentiated plasma cells are prone to a worse prognosis. Thus, the supposition that myeloma initiating/progenitor cells originate from the CD138 negative B cell compartment has been advocated and verified during the past decades. A previous study has revealed that the potential precursor MM cells is most likely situated in the memory B cell compartment and not in the pre-B or immature B cell compartment [40,41,42]. The clonogenic B cells in MM patients have already gone through the phase of somatic hypermutation and antigen selection without intra-clonal heterogeneity, but can still occur during the class switching process [43]. 

Mounting evidence demonstrated that circulating CD19 positive B cells also express surface monoclonal protein and harbor chromosomal abnormalities [44]. These circulating clonotypic B cells can be detected in the majority of MM patients [41]. Compared with the normal donors, MM patients have a larger pool of circulating late-stage CD19^+^ B cells. Moreover, the circulating clonotypic CD19^+^ B cells exhibit resistance to the conventional chemotherapy regimens. Their number can even be found to increase in granulocyte colony-stimulating factor (G-CSF)-mobilized blood of individual patients [40,45,46]. Thus, these postulated progenitor cells of MM consistently act as a therapy-resistant tumor reservoir that drives recurrence. The high frequency of the circulating clonotypic B cells and their resistance to chemotherapy indicate the importance in eradicating this potential stem population and strictly monitoring in the clinical course. Consistently, Hansmann et al. combined the multi-high throughput methods, including multi-parameter FACS single-cell sorting and high-efficiency single-cell immunoglobulin sequencing to trace B cell lineage clones across B cell development stages in MM [47]. They found that clonal MM cells in the bone marrow are not confined to plasma cell compartments but extend to low frequencies of normal phenotype B-lineage cells. The therapeutic success of anti-CD19 chimeric antigen receptor (CAR)-T cells in some MM patients suggests that phenotypically clonal B cells may be clinically relevant therapeutic targets for some selected cases [48,49,50,51]. 

In addition to the widely known marker of CD19, CD24 is also one of the biomarkers in B cells with changed expression levels during the B cell development [52]. CD24 is highly expressed in multiple tumor cells [53]. In particular, CD24 positiveness is identified as a cancer stem cell marker for various types of cancer, such as gastric cancer, colorectal cancer, cervical cancer, etc. [54,55,56]. Interestingly, CD24 was identified as a surface marker for the tumor-initiating cells in MM as well. CD24 is highly expressed in the MM side population (SP) cells, which are supposed to be enriched with progenitor cells in hematological stem cell research and in diverse types of malignancy [57,58,59,60]. Only 10 CD24^+^ MM cells could develop plasmacytomas in vivo [61]. To be noticed, CD24 can also be a promising target for cancer immunotherapy. The interaction of CD24-Siglec-10 is a ‘don’t eat me’ signal for tumor cells [62]. Thus, CD24 targeted therapy might be more effective due to the dual function in eliminating progenitor cells directly and activating the immune cells. However, some evidence has also suggested that CD138^+^CD19^−^ plasmablasts/PCs are the putative MM stem cell compartments [63,64]. Accordingly, the phenotypic characteristics of myeloma initiating cells still remain elusive.

Self-renewal is one of the hallmarks for tumor-initiating cells (TICs). Cell purification using distinct surface markers followed by transplantation in immunocompromised mice has been used as a gold-standard method to test functional TICs populations in diverse malignancies. In myeloma study, due to the high dependency of MM cells on immune microenvironment, it is a challenge to perform the self-renewal experiment of tumor cells in vivo. Some studies pointed out that CD138^−^ B cells in MM patients have the robust ability to replicate and differentiate into the malignant CD138^+^ plasma population in vitro and in vivo [65,66]. But we also noticed that Guikema et al. found that the myeloma clonotypic B cells are hampered in their ability to undergo B cell differentiation in vitro [67]. However, the in vitro B cell differentiation culturing system in this study is stimulated by IL-10 and IL-2. The activation and differentiation signaling might not be sufficient to induce the clonotypic B cells to differentiate into Ig-secreting cells. Therefore, the in vivo differentiation studies of CD138^−^ B cells from MM patients are more convincing in the hypothesis of the TICs populations in CD138^−^ populations in MM patients. 

Although most current studies posit that the origin of MM TICs within the B cell compartment, a comprehensive understanding of their precise phenotypic and biological attributes remains incomplete. As such, a minor TIC population in the B cell compartment is supposed to harbor some unique features at single-cell resolution. Leveraging novel technologies holds the promise to unravel these specific features, offering potential therapeutic targets that may contribute to achieving the ultimate goal of curing the disease and preventing relapse.

**Table 1 cancers-16-00498-t001:** An overview of the studies in MM origination.

Hypothesis That MM Originates from B Cell Compartment			
**Reference**	**Year**	**Methodology**	**Phenotypic Marker**
Bakkus et al. [43]	1994	Immunoglobulin sequence detection	CD19^+^HLA class II^+^IgM^+^
Bergsagel et al. [45]	1995	Immunoglobulin sequence detection	CD38^+^CD19^+^CD56^+^
Szczepek et al. [46,68]	1997, 1998	Immunoglobulin sequence detection by in situ reverse transcription-PCR	CD34^+^CD19^+^
Pilarski et al. [44,69]	2000, 2002	Immunoglobulin sequence detection by in situ reverse transcription-PCR, in vivo clonogenic assay	CD34^+^CD19^+^
Rasmussen et al. [41]	2004	Immunoglobulin gene rearrangement detection	CD38^−^CD19^+^CD27^+^
Matsui et al. [65]	2004	In vitro and in vivo clonogenic assay	CD138^−^
Matsui et al. [40]	2008	In vitro and in vivo clonogenic assay, immunoglobulin gene rearrangement detection	CD138^−^CD20^+^CD27^+^
Boucher et al. [70]	2012	Multicolor flow cytometry, immunoglobulin sequence detection by PCR, in vitro clonogenic assay	CD34^+/−^CD19^+^CD117^+^Survivin^+^Notch^+^
Hansmann et al. [47]	2017	FACS index sorting, single-cell immunoglobulin sequencing	CD19^+^CD20^+^CD45^+^
Kellner et al. [66]	2019	Transgenic mouse model	B220^+^CD19^+^IgM^−^IgD^−^CD138^−^CD80^+^sIgG^−^AA4.1^+^FSC^hi^
Gao et al. [61]	2020	In vitro and in vivo clonogenic assay	CD24^+^
**Hypothesis that MM originates from the plasma cell compartment**			
Kim et al. [64]	2012	In vivo clonogenic assay	CD19^−^CD45^low/−^CD38^high^/CD138^+^
Hosen et al. [63]	2012	In vitro and in vivo clonogenic assay	CD138^−^CD19^−^CD38^++^

### 3.2. Novel Insights into Myeloma Initiation

The integrated multi-omics sequencing data offer a comprehensive analysis to resolve the underlying mechanisms pertaining to malignant transformation of normal PCs to malignant PCs (Table 2). 

Frede et al. performed full-length scRNA-seq and scATAC-seq and identified lineage infidelity and plasticity in myeloma cells. Myeloma cells appeared to be more immature compared to normal plasma cells. Gene signatures of progenitor populations of the lymphoid or entirely hematopoietic precursors were expressed in myeloma cells. Myeloma cells were predicted to have a higher developmental potential. scATAC-seq data further provided evidence that it is enhancer activation that results in the expression of genes which are not normally expressed in MM lineage, ultimately leading to lineage infidelity in MM [74]. Thus, the activated enhancer might endow myeloma cells with the same ability as TICs, which have more potential to develop. Alameda et al. built up a transcriptional atlas of normal PCs development in order to identify the mechanisms of malignant transformation in MM. By integrating bulk and single-cell transcriptional sequencing data, they revealed that PCs of MM are transcriptionally closer to PB-PCs and newborn BM-PCs which are featured alongside the expression of CD39. Patients with more immature transcriptional programs of PCs had inferior survival [85]. In another study, Liang et al. successfully identified that the shared mutation of IFITM2 and ANK1 occur in the early stage of malignant origin to mediate the malignant transformation in MM by combining single-molecule long-read genome sequencing with scRNA-seq [86].

The B cell to plasma cell development involves several main physiologic processes: V(D)J-recombination, affinity maturation, somatic hypermutation, and class-switch recombination. Such physiologic processes could lead to chromosomal DNA deletion and recombination. Once the mechanism of splicing and reassembling DNA occurs in error, it can lead to aberrant chromosomal translocations involving the IGH chain locus at chromosome 14q32.33. If the specific oncogenes are involved in the translocation, they can fall under the control of the IgH enhancer. The enhanced activation of the partner oncogenes could endow cells with growth advantage and give rise to the development of pathological states. Hereby, chromosomal instability (CIN) is a major driver of the MM complex genomic landscape. The genomic landscape is characterized by chromosomal copy number alterations, structural variations, and known cancer driver mutations. Hyperdiploidy and IGH translocations are conventionally regarded as the primary genetic events in myeloma. Consistently, these genomic abnormalities can already be detected at the premalignant stages. Different types of CIN are closely related. The presence of numerical CIN or structural CIN can be induced or accelerated mutually, and these give rise to the increased tendency of new mutations [87]. As such, investigations into the genomic alterations in the MM development are critical for understanding myelomagenesis.

Although the structural variation (SV) landscape of MM is characterized by a lower SV burden and less genomic complexity than in many solid tumors, SVs generally play a crucial role in shaping the genome of MM patients [88]. SVs, including chromothripsis, templated insertions, and chromoplexy, are identified as major drivers for MM evolution, occurring early from the WGS data of sequential MM genomes [89]. These SV classes simultaneously deregulate multiple driver genes as part of a single event, demonstrating that the full genomic landscape of MM can be acquired through a few key events during the tumor evolutionary history [88]. 

The occurrence of SVs in tumor cells appears to be heavily influenced by the 3D genome. The 3D organization of the genome is dynamically regulated in biological processes, and also displays alterations in the physiological state. The 3D organization of the genome and cancer genome alterations reciprocally influence each other. Therefore, deciphering the spatial disorganization of the cancer genome would help to understand the molecular mechanisms of myelomagenesis at a higher resolution. Vice versa, the 3D organization of the genome also exhibits substantial and dynamic heterogeneity shaped by SVs. A recent study has taken an initial step to integrate Hi-C, WGS, and RNA-seq to uncover the correlation between 3D genome architecture and genomic variation and gene expression in MM cell lines. Compared to normal B cells, the number of TADs increases by 25% in MM, and the average size of TADs is smaller. About 20% of genomic regions switch their chromatin A/B compartment types in MM cells [90]. Therefore, the chromosomal spatial alterations might provide more insights into the initiation of MM at a fine scale. 

Alternative splicing (AS) is a critical post-transcriptional event and plays a pivotal role in cellular processes of malignancies, including tumorigenesis, proliferation, invasion, metastasis, and resistance to drugs. AS is not reflected in overall gene expression levels, but the combination of splice sites within a pre-mRNA by AS is able to generate variably spliced mRNAs and to further drive proteome diversity. Splicing factors appear to be dysregulated in MM cells compared to normal plasma cells and further drive the aberrant AS [91,92]. AS targeting DNA replication, cell cycle, and apoptosis pathways has been observed in myeloma patients by deep RNA-sequencing. The non-homologous end joining pathway is an independent factor highly associated with splicing frequency as well as an increased number of structural variants. However, the splicing pattern has not been comprehensively understood in myeloma initiation and progression [92]. 

When investigating the mutational landscape of MM by WGS and WES, the chronological activity of each mutational signature can be reconstructed. The timing of mutational processes demonstrates that the initial transformation of a germinal center B cell mostly occurred during the second–third decades of life. The pre-malignant MM cells behave similar to memory B cells, capable of re-entering the germinal center to undergo clonal expansion decades before MM diagnosis. The first chromosomal multi-gain event is estimated to occur before 30 years of age in the majority of MM patients [93]. 

Dysregulated metabolism is one of the hallmarks of cancer. A distinct metabolic profile has been identified in MGUS/MM patients [94]. Single-cell proteomics is a field in its infancy [95]. Single-cell proteomics holds the potential to provide comprehensive insights into the pathogenesis of MM. The initiation of malignant transformation evolves many factors. Previous research could not include the multi-dimensional factors in one study due to the technological limitations. Therefore, the deeper insights into the mechanisms relevant to the malignant transformation could be underscored by virtue of the improved single-cell sequencing technology. 

## 4. Clonal Evolution

### 4.1. Clonal Evolution during MM Course

Common initiating events that are almost always present in premalignant precursors encompass IGH translocations and hyperdiploidy [96]. Secondary cytogenetic hits could be observed when progressing to MM accompanying with the treatment and recurrent disease relapse. Collectively, the natural history of disease transformation from MGUS to overt MM have inferred that clonal evolution is a crucial feature for the disease progression in MM. Intercepting the progression of clonal evolution could potentially impede the transformation from the premalignant stage to MM with the appropriate clinical interventions. 

The clonal evolution patterns include stable evolution, branch evolution, and linear evolution [97]. When experiencing relapses and progression, clonal evolution can be observed longitudinally in MM patients. Bulk investigation into the clonal evolution of tumor cells is inadequate for elucidating the principles governing clonal evolution and the driving mechanisms behind it. The higher resolution research into the clonal evolution could intensify the understanding of disease progression and develop the precision intervention to prevent the occurrence of malignant transformation [97]. Conventional fluorescence in situ hybridization (FISH) is the gold standard to evaluate the cytogenetic risk factors and to guide the risk stratification in MM. Quantitative multigene fluorescence in situ hybridization (QM-FISH) enables the detection of multiple chromosomal changes synchronously at the single-cell level. Nowadays, single-cell CNA sequencing definitely improves the detection level [98]. With single-cell genomic analysis, CNA subclonality is a general process in MM and in premalignant stages as well, but it is not detectable by the current routine assessment [99]. 

Clonal evolution of tumor cells is also accompanied by transcriptomic changes. Increased activation of proliferative and metabolic signal pathways is identified during disease progression, with the new clone acquiring new myeloma driver mutations or CNAs [100]. Clonal evolution could drive tumor progression, dissemination, and relapse in MM. Tumor cells which experience clonal selection are found to evolve quickly, compete with other clones, intravasate into the tumor vasculature, adapt to the environment, and ultimately extravasate for further colonization [97]. 

Genomic heterogeneity is magnified during clonal evolution. A range of clones compete for dominance and adapt to the local environment. WGS and WES provide a genomic variance profile during the clonal evolution process. Serial WES analyses of paired MGUS-MM or SMM-MM samples demonstrated that most somatic mutations are already present at the premalignant stages of MGUS and SMM [101]. The genetic complexity increases as MGUS progresses to MM, and the mutation load is associated with poor prognosis [96]. Myeloma subclones exhibit distinct survival properties facing to the treatment pressure [97,102]. In relapsed MM patients, the mutational landscape of tumors is heavily shaped by exposure to chemotherapeutic agents [103,104]. The chemotherapeutic agents play a key role in accelerating genomic complexity and contributing to a considerable proportion (>20%) of the nonsynonymous mutations. A single propagating myeloma cell surviving after chemotherapy exposure could take the clonal dominance, expand, and accelerate the disease seeding [104]. 

Notably, the clonal evolution pattern is of prognostic significance. The significance of baseline cytogenetic aberrations is the most pronounced at diagnosis and attenuated over time after treatment. Newly acquired cytogenetic aberrations during the 3 years after diagnosis is correlated with the increased mortality [105]. Patients with clonal stabilization had a better overall survival. Also, clonal evolution with the presence of high-risk cytogenetic aberrations at relapse will greatly affect the survival after relapse [98].

In addition to detecting clonal evolution by cytogenetic aberrations, a recent study incorporated phenotypic patterns by mass cytometry for myeloma subclonal analysis. The correlation existed between specific phenotypic cluster and treatment response and overall survival independent of genetic aberrations or patient demographic variables. Thus, the clonal composition of myeloma is very complex [106]. The current detection of clonal evolution is mainly decided by genomic information. It will be worth exploring the significance of integrating genomics, epigenomics, and protein profiling together in myeloma evolution.

### 4.2. Novel Insights into Alterations of Malignant Plasma Cells in the Disease Evolution

The progression of multiple myeloma follows a paradigm from the able-to-be-identified pre-neoplastic condition to the overt neoplastic condition. Diverse disease status is distinguished by the physical symptoms and clinical examination. Nevertheless, the underlying molecular signatures and mechanisms still need to be further elucidated. Although plasma cells from MM patients displayed highly unique transcriptional programs, they also shared overexpressed common MM oncogenic genes, such as CCND1, CCND2, and FGFR3. With the advantage of scRNA-seq technology, more putative MM-specific oncogenic genes were identified, such as LAMP5, CDR1, and WFDC2 [80]. Additionally, the NF-kB pathway is of great importance in MM. Mutations in 11 members of the NF-kB pathway have been indicated by WGS [107]. 

Jang and colleagues examined molecular heterogeneity among different stages of disease progression including MGUS, SMM, NDMM, and RRMM by scRNA-seq. Plasma cells from MGUS patients have the unique characteristics which could distinguish from plasma cells from other disease stage. Additionally, plasma cells with different cytogenetic abnormalities present as distinct clusters, suggesting a close correlation between cytogenetic abnormalities and cellular properties [78]. Cytogenetic variations apparently play a prominent role in the disease progression. 

One of the most typical symptoms of myeloma is osteolytic lesions (OL). Discrepancies of PC between bone marrow and osteolytic lesions cause the spatial heterogeneity in MM. PCs from BM and paired imaging-guided biopsies of OL exactly showed transcriptomic discrepancies. Genes associated with myeloma bone disease, such as DKK1, HGF, and TIMP-1, were found to be upregulated in PC from OL. The downregulation of genes, like JUN/FOS, DUSP1, and HBB which has been connected to extramedullary spread of MM, was also found in PC from OL. For one patient with a PC tumor with para-medullary spread, the most significant differences could be discovered between PC from BM and OL by single-cell sequencing. A small cluster of PC with low AZGP1 expression could be identified in BM and the respective cluster also connected to upregulation in the epithelial–mesenchymal transition (EMT) and the downregulation of proliferation and oxidative phosphorylation. This small PC cluster might have occurred latest in the development and given rise to the OL [73]. 

For most conditions, MM cells are confined within the bone marrow and are highly dependent on the microenvironment to survive. When clonal plasma cells are able to grow outside of bone marrow and arrive at anatomic sites distant from bone marrow or periphery, it is referred to as extramedullary multiple myeloma disease (EMD) or secondary plasma cell leukemia (sPCL). EMD or sPCL can be present at relapse. Despite high-dose chemotherapy with autologous stem cell transplantation and novel therapeutic agents, prognoses of MM patients with EMD remains poor. The underlying mechanisms for extramedullary invasion of MM cells are not completely understood. The known factors effecting the dissemination of MM cells include (1) downregulation of chemokine receptors, adhesion molecules, and tetraspanins, (2) molecular pathway activation, such as proliferation/cell-cycle progression, glycolysis, oxidative phosphorylation, proteasome, and antigen presentation [71], and (3) upregulation of angiogenic cytokines [108]. Apart from the above mechanisms, by virtue of the advantage of single-cell sequencing technology in detecting rare cells, recent research demonstrated that CXCL12 was identified as the significant upregulated gene in circulating plasma cells from patient with EMD among multiple chemokines. More importantly, key molecules related to chemokine secretion, including VAMPs, Rabs, and syntaxins, were simultaneously upregulated in cPCs in patients with EMD. A possible self-feed loop machinery for chemokines may release myeloma cells from receptor-dependent bone marrow retention into the circulation [109]. Therefore, during disease evolution, tumor cells can acquire novel genomic alterations, become more invasive, extravasate from the bone marrow, and finally result in the end-stage of MM. It is essential to understand the evolution mechanisms of MM cells and to circumvent the progression. The longitudinal sample collection in compliance with the research poses inherent challenges, thus necessitating more in-depth and elaborate methods to integrate the data for analysis. 

## 5. Drug Resistance and Potential Therapeutic Targets

The current MM therapy options include proteasome inhibitors, immunomodulatory agents, glucocorticoids, chemotherapeutic agents, and immunotherapy. Most patients initially respond to anti-myeloma treatment, but they eventually become non-responsive to subsequent treatment. Additionally, precision treatment of specific mutated MM cells often fails to elicit substantial treatment responses, despite the fact that the mutation has occurred at the initial time [110]. Drug resistance and recurrent relapse have always been the main cause for the reduced survival and heavy burden on both the economical and mental status of myeloma patients. 

The transcriptional alterations can actually take place prior to the detectable outgrowth of drug-resistant clones, accompanied with widespread enhancer remodeling [110]. Therefore, it is possible to realize the prediction of a response to treatment by molecular profiling of tumor cells [111]. The KYDAR (DARA-KRD) clinical trial (ClinicalTrials.gov ID: NCT04065789) is designed for relapsed/refractory patients who have received a bortezomib-containing regimen before but failed to respond or progressed early. Researchers from the Weizmann Institute integrated single-cell transcriptome technology with this clinical study and proposed a 66-gene bortezomib-resistant module which performed well in distinguishing primary refractory malignant PC, including STMN1, COX6C, SOD1, PPIA, and PSMB4. The resistant gene module was characterized by perturbation in proteasome machinery, mitochondrial stress, ER stress, and the UPR pathway [77]. 

The acquisition of drug resistance may occur in multiple stages. The existence of CSCs or TICs are suggested to lead to drug resistance in tumors. The autocrine sonic hedgehog (Hh) signaling pathway is closely associated with cell stemness and chemoresistance in several types of cancer [112]. Consistently, the activation of the Hh pathway can be seen in the bortezomib-resistant (BR) cells. Clustering and trajectory analysis demonstrated that BR cells followed a different differentiation trajectory from wild-type (WT) cells. The resistant clones are either the stem-like ones initially in MM or are induced by the treatment [113].

When referring to the incidence of drug resistance, measurable residual disease (MRD) cells are the considered as the important research samples. No unifying genetic abnormalities are identified in MRD. Therefore, MRD are more likely to emerge from the selection of tumor cells that transcriptionally evolve to resist therapy under the treatment pressure. MRD cells are indicated to be in a quiescent and metabolically inactive transcriptional state as well. In addition to the general hallmark resistant pathway related to proteasome activity, impaired mitochondrial translation and metabolic gene expression pathways can be observed in MRD cells [110,114].

In the MM animal model, tumor cells in a dormant state are also validated to be able to evade drug cytotoxicity and lead to relapse when they are reactivated. The endosteal niche cells, such as osteoblast lineage cells, induce a myeloid transcriptome signature in dormant MM cells by cell contact. Moreover, the myeloid transcriptome signature disappeared in reactivated MM cells, indicating a niche-dependent mechanism for the maintenance of drug resistance. Inhibition of these signature genes releases dormant cells and converts them into activated ones. These dormant cells are akin to the stem-cell population in MM, characterized by its resting state, and rely on the surrounding microenvironment. Thus, it might be an alternative strategy to eradicate dormant tumor cells by revitalizing the dormant tumor cells in a first step and combining this with sequential chemotherapy [115]. Later in the course of MM, genetic abnormalities and epigenetic aberrations, mainly affecting the patterns of DNA methylation and histone modifications of genes, are well known to be involved in the resistance mechanism [110]. Additionally, the surrounding microenvironment and the extracellular vesicles also play a role in MM drug resistance as well. To address the drug resistance problem in MM, the multi-omics sequencing platform has also been used for screening, predicting, and detecting novel drug combination candidates [116]. 

## 6. Impaired Tumor Microenvironment in Multiple Myeloma

The survival of myeloma cells is intricately linked to the supportive nature of their bone marrow niche. Adjacent cells within this microenvironment bolster tumor cells through direct cell-to-cell interactions or the secretion of cytokine factors. Vice versa, myeloma cells actively modulate the microenvironment to create a more favorable survival space. However, the elucidation of the complex interactions between the immune and non-immune compartment has been restricted by the biased selection of cell types or differentiation states for enrichment and analysis [117]. To overcome this limitation, the application of single-cell sequencing and immune repertoire analysis can be instrumental. These technologies allow for an unbiased characterization of the diversity and variation within the immune cell compartment, enabling a comprehensive dissection of connections among different cellular populations.

Immune dysfunction can be observed at different disease stages, even in the early stages, including MGUS, SMM and overt MM [118,119]. At the premalignant stage (MGUS stage), stem-like and marrow-resident features in T cells were observed both at the proteomic level (mass cytometry) as well as at the transcriptomic level (single-cell transcriptome analysis). Increased expression of TCF1 in T cell compartments associated with T cell stemness are also accompanied by increased expression of CXCR4, NR4A2, and CD69, which are related to bone marrow residency [75]. As for myeloid compartments, CD14^+^ monocytes show impaired antigen presentation and confer advantages in proliferation to myeloma cells and suppress T cell activation at the same time. When progression to the overt MM stage occurs, the integral immune profiling of the MM microenvironment is characterized by an increase in NK, T, and CD16^+^ cells, as well as a decrease in plasmacytoid DCs, immature neutrophils, CD14^+^ monocytes, and other progenitor cells. Among T cell subsets, there is a significant upregulation of Tregs and a depletion of memory T cells under disease status [81,120]. Although the general abnormalities of immune cells can be observed in most MM patients, there still exist multiple impact factors on the immune cells. Our previous study confirmed the dynamic alteration of immune cells along with the elevated infiltration of tumor cells in bone marrow [118]. Immune cells were activated in patients with low infiltration of MM cells, while they were suppressed with elevated infiltration of MM cells. The differentiation status, expression of immune modulators, and metabolism status changed with the infiltration of MM cells. Disordered amino acid metabolism and lipid metabolism in immune cells promote the dysfunction of immune cells and defective immune response in MM. 

The extrinsic features of the bone marrow niche also deliver positive signals to support extramedullary myeloma progression. Dysregulated immune molecules on MM cells suggested a suppressive effect on immune cells, including dynamic regulation of MHC class I gene expression and LILRB1/4-mediated NK inhibition, the TRAIL/TRAIL receptor, and HIF1A/PD-L1, which induce the immune evasion of MM cells [71].

Apart from conventional immune cells, mesenchymal stromal cells also play a critical role in modulating the bone marrow niche and nourishing the malignant cells [121,122]. In the non-hematopoietic microenvironment, myeloma-specific mesenchymal stromal cells with the inflammatory phenotype (iMSC) were enriched in the inflammation-related pathway “TNFɑ signaling via NFκB”. A few types of immune cells were identified as potential sources of TNF which can interact with iMSCs, including interferon (IFN)-responsive effector T cells and CD8^+^ stem cell memory T cells (Tscm). Furthermore, the activation of iMSCs by TNF signal is poised for immune cell modulation and tumor cell proliferation. Strikingly, this kind inflammatory bone marrow niche cannot be reversed by the current antitumor therapy even when reaching MRD negativity [79]. 

The in-depth investigation of the complex immune compartment is compatible with the single-cell resolution methodology. More elaborate studies of the bone marrow niche including tumor and non-tumor compartments will help to define the molecular basis for TME-targeted therapeutic approaches and provide clinical benefits.

## 7. Conclusions

The multi-step progression of MM presents opportunities to curb the disease progression by targeting the pivotal pathways. Unraveling the origin of myeloma cells and the evolution mechanisms will help to realize precision intervention and precision medicine strategies. Genomic instability and aberrant tumor microenvironments endeavor together to induce the progression, dissemination, and evolution of MM. Vast heterogeneity occurs among the patients and across the patients at different stages of MM. 

Multi-omics sequencing technologies have greatly enhanced the knowledge about the disease, from diagnosis to treatment. However, the initiation, evolution, and progression of MM are not fully understood. The contemporary surge in sequencing data poses a challenge due to the diverse methodologies and platforms employed in various studies. As a result, there is a need to continually design and enhance computational algorithms for effective data integration and analysis. Simultaneously, the field of multi-omics sequencing is advocated and evolving. The integration of the multi-omics sequencing data is important for providing the multi-dimensional information of the specific cell population. In addition to the data analysis, the application of multi-omics sequencing in one single cell also faces technological limitations. 

At this time, the prevailing genomics and epigenomics profiling of MM have facilitated in-depth analyses of the genome of MM cells. However, the current proteomic profiling of MM is predominantly restricted to the cellular surface proteome. MM is characterized by its aberrant secretion of immunoglobulin. The protein synthesis and degradation process are active and complex. Thus, the breakthroughs in proteomic profiling have the potential to enhance the understanding of the disease, such as MM clonal definition and tracing, early diagnosis, and novel therapeutic screening. 

## Figures and Tables

**Figure 1 cancers-16-00498-f001:**
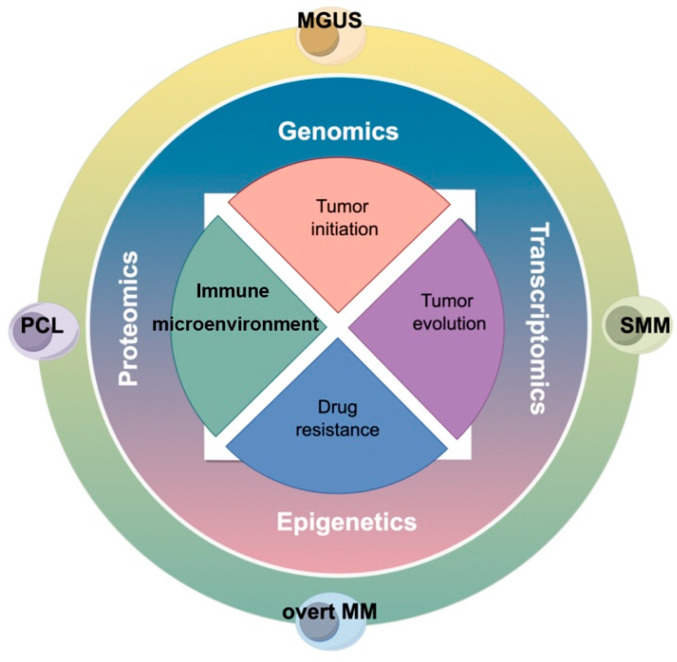
The graphical overview of the review.

**Table 2 cancers-16-00498-t002:** Multifaceted single-cell research using different methodologies in MM.

Reference	Sample Type	Methodology	Major Findings
Ryu et al. [71]	BM or extra-medullary samples of MM (n = 15)	Whole-transcriptome sequencing for CD138^−^ negative cells, full-length scRNA sequencing for CD138^−^ positive cells	Transcriptional programs associated with extramedullary progression support autonomous cell proliferation and immune invasion.
Liu et al. [72]	Longitudinal BM samples at different disease stages (n = 14)	10× Genomics scRNA sequencing, 10×WGS, CyTOF	Presented co-evolution maps of tumor and immune cells during MM disease development.
Merz et al. [73]	Paired BM and biopsies of osteolytic lesions from MM patients (n = 10)	10× Genomics scRNA sequencing, WES	Identified spatial transcriptional changes in MM.
Frede et al. [74]	BM and PB of RRMM (n = 8) at diagnosis and following treatment and HDs (n = 2)	Full-length scRNA sequencing, scATAC sequencing	Differential regulon usage and enhancer rewiring underlies distinct transcriptional states of cancer cells. Transcriptional reprogramming and differential enhancer recruitment by treatment pressure underlies drug resistance.
Bailur et al. [75]	BM immune cells from HDs (n = 12) and MGUS/MM patients (n = 26)	10× Genomics scRNA sequencing, CyTOF	Identified early alterations in stem-like /marrow-resident T cells and innate and myeloid cells in the MGUS stage.
Lohr et al. [76]	Single myeloma circulating tumor cells and BM-derived MM cells from MM patients (n = 10)	scRNA sequencing, scDNA sequencing	Single-cell sequencing enables genetic interrogation of circulating tumor cells with greater sensitivity.
Cohen et al. [77]	BM PCs from 60 individuals: controls (n = 11), NDMM (n = 15) and PRMM patients (n = 41)	Single-cell MARS-seq	Identified drug-resistant pathways and therapeutic targets in high-resistant MM patients.
Jang et al. [78]	BM PCs from 15 patients at different stages of disease progression: MGUS (n = 3), SMM (n = 4), NDMM (n = 5), RRMM (n = 3)	Single-cell MAPRSeq	Identified gene expression signatures and molecular pathways during disease progression.
Jong et al. [79]	BM PCs and immune cells from HDs (n = 7) and NDMMs (n = 13)	10× Genomics scRNA sequencing	Inflammatory mesenchymal stromal cells are involved in tumor survival and immune modulation and lead to disease persistence.
Ledergor et al. [80]	BM PCs from 40 individuals: HDs (n = 11), MGUS (n = 7), SMM (n = 6), NDMM (n = 12) and AL (n = 4)	Single-cell MARS-seq	Dissected high interindividual heterogeneity and molecular characterization of tumor cells in symptomatic and asymptomatic patients.
Zavidij et al. [81]	BM immune cells from MGUS (n = 5), low-risk SMM (n = 3), high-risk (n = 8), NDMM (n = 7) and HDs (n = 9).	10× Genomics scRNA sequencing	Transcriptional and compositional alterations in the microenvironment occur early in precursor stages of MM.
Boiarsky et al. [82]	BM PCs from HDs (n = 9), MGUS (n = 6), SMM (n = 12), and NDMM (n = 8)	10× Genomics scRNA sequencing	Characterized transcriptomic alterations of tumor cells at the earliest stages of MM.
Dang et al. [83]	BM samples from HDs (n= 4), MGUS (n = 21), SMM (n = 32), NDMM (n = 7), and RRMM (n = 1)	10× Genomics scRNA and scBCR sequencing	A comprehensive analysis of the clonotypic and transcriptional evolution of tumor cells and microenvironment from precursor disease to MM.
Friedrich et al. [84]	BM immune cells from HDs (n = 30), NDMM (n = 7), and RRMM (n = 18) receiving BCMA×CD3 bispecific TCE monotherapy	10× Genomics scRNA and scTCR sequencing	A comprehensive longitudinal profiling of the BM T cell repertoire and its response to TCE treatment in MM patients.

BM: bone marrow; PB: peripheral blood; NDMM: newly diagnosed MM patients; RRMM: relapsed/refractory MM; PRMM: primary refractory MM patients; MGUS: monoclonal gammopathy of undetermined significance; AL: amyloidosis; PC: plasma cells; WGS: whole-genome sequencing; WES: whole-exome sequencing; CyTOF: cytometry by time-of-flight; TCE: T cell engagers.

## Data Availability

The data can be shared up on request.

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
