# Peer review of "Novel Insights into the Initiation, Evolution, and Progression of Multiple Myeloma by Multi-Omics Investigation"

_cancers, 2024, doi:10.3390/cancers16030498_

Round 1

Reviewer 1 Report

Comments and Suggestions for Authors

To the authors

The authors explore the evolutionary history of multiple myeloma (MM), emphasizing malignant transformation, pre-malignant stages, and the development of aggressive, treatment-resistant forms. Despite a decade of efforts to identify therapeutic targets, MM remains largely incurable, leading to frequent relapses and treatment resistance.

The authors suggest that tumor-initiating cell populations contribute to recurrent relapses, and clonal evolution of tumor cells plays a key role in disease progression. They highlight the intricate role of the tumor microenvironment in supporting tumor cell evolution.

Utilizing emerging single-cell multi-omics research, the authors provide novel insights into MM, discussing tumor cell initiation, clonal evolution, drug resistance, and the tumor microenvironment. They propose that these insights offer a promising strategy to understand pivotal mechanisms in MM progression, potentially leading to more effective and precise treatment approaches.

  1. Clarity of Premalignant Stage Intervention Importance:
    • Strengths: The authors effectively stress the significance of intervening at premalignant stages, recognizing the challenge of predicting MGUS/SMM progression to MM.
    • Limitations: The text lacks statistical data or recent studies supporting the assertion that intervention at premalignant stages can significantly impact MM outcomes. Incorporating recent clinical evidence could strengthen this point.
  2. Debatability of Cancer Stem Cells (CSCs) in MM:
    • Strengths: The authors acknowledge the ongoing debate regarding the existence of CSCs in MM.
    • Limitations: While the debate is mentioned, the authors could provide a concise overview of contrasting studies and perspectives in the field, offering readers a clearer understanding of the current state of research on this topic.
  3. Complexity of Malignant Transformation and Clonal Evolution:
    • Strengths: The discussion on the multifaceted process of malignant transformation and clonal evolution is comprehensive.
    • Limitations: The section lacks specific examples or case studies to illustrate the points made. Integrating real-world examples or recent discoveries in this area would enhance the impact of the discussion.
  4. Phenotypic Characteristics of Tumor-Initiating Cells:
    • Strengths: The authors present a detailed exploration of phenotypic characteristics, particularly the role of CD19+ B cells and CD24 as potential markers.
    • Limitations: The information on CD24 is somewhat scattered. Organizing this information into a cohesive narrative and providing a comparative analysis with other potential markers could enhance clarity and reader understanding.
  5. Controversy Surrounding Self-Renewal Abilities:
    • Strengths: The authors present conflicting findings on the self-renewal abilities of CD138- B cells, acknowledging the controversy in the field.
    • Limitations: To provide a balanced view, the authors should consider summarizing the conflicting findings and proposing potential explanations for the discrepancies.
  6. Importance of Advanced Methodologies:
    • Strengths: The authors highlight the importance of advanced methodologies like high-throughput methods and single-cell sequencing.
    • Limitations: It would be beneficial to mention specific advancements in these methodologies and potential challenges researchers face when applying them to MM studies.
  7. Single-Cell Sequencing Advancements:
    • Strengths: The discussion on the higher resolution clonal map provided by single-cell sequencing is insightful.
    • Limitations: Incorporating recent studies or comparative analyses with traditional sequencing methods could reinforce the advantages of single-cell sequencing.

Finally, the authors mentioned CSC in multiple myeloma, this reviewer personally misses some insights regarding autophagy.  Autophagy plays a multifaceted role in MM, influencing CSC survival, self-renewal, therapy resistance, and interactions with the tumor microenvironment. CSCs, known for their resistance to conventional treatments, may rely on autophagy for survival. Inhibiting autophagy could potentially sensitize CSCs to therapy. Additionally, the crosstalk between autophagy in cancer cells and the tumor microenvironment creates a niche supporting CSC maintenance. Autophagy's impact on metabolic regulation, genomic stability, and cellular differentiation further complicates its role in MM. The article highlights the need for continued research to unravel these complexities and develop targeted therapies for more effective MM treatment. Please refer to PMID: 26374531 and PMID: 32085480 and expand

Overall Recommendations for Improvement:

  • The inclusion of recent statistical data, case studies, and clinical evidence could strengthen various arguments.
  • A more organized presentation, particularly in the CD24 section, would enhance readability.
  • A concise summary of conflicting findings and potential explanations would enrich the discussion.
  • Incorporating recent advancements and challenges in methodologies would provide a more current perspective.
  • Integrating comparative analyses and examples could enhance the overall impact of the review.

To the editors

In their exploration of the evolutionary history of multiple myeloma (MM), the authors delve into the intricate processes involving malignant transformation, progression to pre-malignant stages, and eventual development into aggressive and resistant forms of the disease. Despite a decade of concerted efforts to pinpoint therapeutic targets for MM, the authors note the persistent challenge of its incurability, with patients frequently experiencing relapses and developing resistance to treatment.

The authors propose that tumor-initiating cell populations may contribute to recurrent relapses in hematological malignancies, including MM. They identify clonal evolution of tumor cells as a significant factor in disease progression, wherein the heterogeneous MM cell clones exhibit varying responses to treatment, allowing more aggressive populations to survive and evolve.

The tumor microenvironment is recognized as a complex ecosystem playing multifaceted roles in supporting the evolution of tumor cells. The authors highlight the importance of emerging multi-omics research at single-cell resolution, which enables an integrative and comprehensive profiling of both tumor cells and the microenvironment. This approach enhances the understanding of the biological features of MM.

In their review, the authors aim to discuss novel insights into tumor cell initiation, clonal evolution, drug resistance, and the tumor microenvironment in MM, as revealed by the advancements in multi-omics investigation. They assert that these data provide a promising strategy for unraveling pivotal mechanisms in MM progression, offering the potential for holistic and precise improvements in treatment strategies.

In this introduction, the authors discuss the multi-step process of multiple myeloma (MM) evolution from premalignant phases (MGUS and SMM) to symptomatic MM. They underscore the challenge of predicting progression from MGUS/SMM to MM, emphasizing the importance of intervening at the premalignant stages. The concept of cancer stem cells (CSCs) is explored in the context of MM, noting the debate over their existence. The authors delve into the multistep and multifaceted process of malignant transformation, driven by clonal evolution, genetic lesions, and interactions with the tumor microenvironment.

The authors review the exploration into tumor-initiating cells in MM, focusing on phenotypic characteristics. They discuss the speculation that myeloma-initiating/progenitor cells contribute to disease recurrence. The role of circulating CD19+ B cells with chromosomal abnormalities and resistance to chemotherapy is highlighted, suggesting their importance as a therapy-resistant tumor reservoir. The authors examine the controversy surrounding the self-renewal ability of CD138- B cells in MM and the significance of side population (SP) cells with stem-like phenotypes.

The phenotypic characteristics of myeloma-initiating cells are explored, including the potential relevance of CD24 as a marker. The authors point out the importance of advanced methodologies, such as high-throughput methods and single-cell sequencing, in investigating minor progenitor populations. They emphasize the need for a more comprehensive understanding of the phenotypic characteristics of myeloma-initiating cells to identify therapeutic targets and improve treatment outcomes.

In the final section, the authors highlight the advancements in single-cell sequencing technology, providing a higher resolution clonal map of MM cells and offering a comprehensive analysis of the mechanisms underlying malignant transformation. They suggest that these novel insights into myeloma initiation could lead to improved therapeutic strategies and ultimately contribute to the goal of curing the disease and preventing relapse.

The provided text is an extensive scientific review of multiple myeloma (MM), covering various aspects of its molecular and cellular biology. To facilitate a peer review, let's break down the text into key points, highlight limitations, and suggest improvements:

1.    Strengths:

    • Comprehensive Literature Review: The review extensively covers recent studies and research findings related to MM, incorporating various techniques such as scRNA-seq, scATAC-seq, WGS, and WES.
    • In-Depth Exploration: The text delves into different aspects of MM progression, including clonal evolution, drug resistance, alterations in the tumor microenvironment, and potential therapeutic targets.

2.    Limitations and Suggestions for Improvement:

·       Clarity and Structure:

    • Issue: The text lacks clear section headers, making it challenging for readers to navigate and identify specific topics.
    • Suggestion: Introduce clear section headers for each major topic (e.g., "Clonal Evolution," "Drug Resistance," "Tumor Microenvironment") to enhance readability.

·       Redundancy:

    • Issue: There's some repetition of information, particularly regarding the techniques used (e.g., scRNA-seq, WGS) and general statements about the importance of understanding MM.
    • Suggestion: Consolidate repetitive information and focus on presenting key findings in a concise manner.

·       Citation Format:

    • Issue: The text lacks a standardized citation format, making it challenging to trace specific references.
    • Suggestion: Adopt a consistent citation style (e.g., APA, MLA) to improve clarity and enable readers to locate the referenced studies easily.

·       In-Text Figures:

    • Issue: The text is dense, and the inclusion of in-text figures or diagrams could aid in visualizing complex concepts.
    • Suggestion: Consider incorporating relevant figures or diagrams to illustrate key processes, pathways, or findings.

·       Interconnection of Ideas:

    • Issue: The transition between different topics could be smoother, and the interconnection of ideas between sections needs improvement.
    • Suggestion: Use transitional sentences or paragraphs to better connect ideas and maintain a logical flow between sections.

·       Discussion of Methodological Limitations:

    • Issue: The review lacks a discussion of potential limitations associated with the methodologies used in the referenced studies.
    • Suggestion: Include a section discussing the limitations of the methods employed in the studies, addressing issues such as sample size, biases, or technological constraints.

·       Conciseness:

    • Issue: The text is quite extensive, and some sections may benefit from concise summarization.
    • Suggestion: Where applicable, provide concise summaries of key findings to maintain reader engagement and clarity.

Incorporating these suggestions should enhance the overall readability, cohesiveness, and impact of the scientific review (please refer to PMID: 32085480 and expand).

The manuscript can be reconsidered after major revisions.

Comments on the Quality of English Language

To the authors

The authors explore the evolutionary history of multiple myeloma (MM), emphasizing malignant transformation, pre-malignant stages, and the development of aggressive, treatment-resistant forms. Despite a decade of efforts to identify therapeutic targets, MM remains largely incurable, leading to frequent relapses and treatment resistance.

The authors suggest that tumor-initiating cell populations contribute to recurrent relapses, and clonal evolution of tumor cells plays a key role in disease progression. They highlight the intricate role of the tumor microenvironment in supporting tumor cell evolution.

Utilizing emerging single-cell multi-omics research, the authors provide novel insights into MM, discussing tumor cell initiation, clonal evolution, drug resistance, and the tumor microenvironment. They propose that these insights offer a promising strategy to understand pivotal mechanisms in MM progression, potentially leading to more effective and precise treatment approaches.

  1. Clarity of Premalignant Stage Intervention Importance:
    • Strengths: The authors effectively stress the significance of intervening at premalignant stages, recognizing the challenge of predicting MGUS/SMM progression to MM.
    • Limitations: The text lacks statistical data or recent studies supporting the assertion that intervention at premalignant stages can significantly impact MM outcomes. Incorporating recent clinical evidence could strengthen this point.
  2. Debatability of Cancer Stem Cells (CSCs) in MM:
    • Strengths: The authors acknowledge the ongoing debate regarding the existence of CSCs in MM.
    • Limitations: While the debate is mentioned, the authors could provide a concise overview of contrasting studies and perspectives in the field, offering readers a clearer understanding of the current state of research on this topic.
  3. Complexity of Malignant Transformation and Clonal Evolution:
    • Strengths: The discussion on the multifaceted process of malignant transformation and clonal evolution is comprehensive.
    • Limitations: The section lacks specific examples or case studies to illustrate the points made. Integrating real-world examples or recent discoveries in this area would enhance the impact of the discussion.
  4. Phenotypic Characteristics of Tumor-Initiating Cells:
    • Strengths: The authors present a detailed exploration of phenotypic characteristics, particularly the role of CD19+ B cells and CD24 as potential markers.
    • Limitations: The information on CD24 is somewhat scattered. Organizing this information into a cohesive narrative and providing a comparative analysis with other potential markers could enhance clarity and reader understanding.
  5. Controversy Surrounding Self-Renewal Abilities:
    • Strengths: The authors present conflicting findings on the self-renewal abilities of CD138- B cells, acknowledging the controversy in the field.
    • Limitations: To provide a balanced view, the authors should consider summarizing the conflicting findings and proposing potential explanations for the discrepancies.
  6. Importance of Advanced Methodologies:
    • Strengths: The authors highlight the importance of advanced methodologies like high-throughput methods and single-cell sequencing.
    • Limitations: It would be beneficial to mention specific advancements in these methodologies and potential challenges researchers face when applying them to MM studies.
  7. Single-Cell Sequencing Advancements:
    • Strengths: The discussion on the higher resolution clonal map provided by single-cell sequencing is insightful.
    • Limitations: Incorporating recent studies or comparative analyses with traditional sequencing methods could reinforce the advantages of single-cell sequencing.

Finally, the authors mentioned CSC in multiple myeloma, this reviewer personally misses some insights regarding autophagy.  Autophagy plays a multifaceted role in MM, influencing CSC survival, self-renewal, therapy resistance, and interactions with the tumor microenvironment. CSCs, known for their resistance to conventional treatments, may rely on autophagy for survival. Inhibiting autophagy could potentially sensitize CSCs to therapy. Additionally, the crosstalk between autophagy in cancer cells and the tumor microenvironment creates a niche supporting CSC maintenance. Autophagy's impact on metabolic regulation, genomic stability, and cellular differentiation further complicates its role in MM. The article highlights the need for continued research to unravel these complexities and develop targeted therapies for more effective MM treatment. Please refer to PMID: 26374531 and PMID: 32085480 and expand

Overall Recommendations for Improvement:

  • The inclusion of recent statistical data, case studies, and clinical evidence could strengthen various arguments.
  • A more organized presentation, particularly in the CD24 section, would enhance readability.
  • A concise summary of conflicting findings and potential explanations would enrich the discussion.
  • Incorporating recent advancements and challenges in methodologies would provide a more current perspective.
  • Integrating comparative analyses and examples could enhance the overall impact of the review.

To the editors

In their exploration of the evolutionary history of multiple myeloma (MM), the authors delve into the intricate processes involving malignant transformation, progression to pre-malignant stages, and eventual development into aggressive and resistant forms of the disease. Despite a decade of concerted efforts to pinpoint therapeutic targets for MM, the authors note the persistent challenge of its incurability, with patients frequently experiencing relapses and developing resistance to treatment.

The authors propose that tumor-initiating cell populations may contribute to recurrent relapses in hematological malignancies, including MM. They identify clonal evolution of tumor cells as a significant factor in disease progression, wherein the heterogeneous MM cell clones exhibit varying responses to treatment, allowing more aggressive populations to survive and evolve.

The tumor microenvironment is recognized as a complex ecosystem playing multifaceted roles in supporting the evolution of tumor cells. The authors highlight the importance of emerging multi-omics research at single-cell resolution, which enables an integrative and comprehensive profiling of both tumor cells and the microenvironment. This approach enhances the understanding of the biological features of MM.

In their review, the authors aim to discuss novel insights into tumor cell initiation, clonal evolution, drug resistance, and the tumor microenvironment in MM, as revealed by the advancements in multi-omics investigation. They assert that these data provide a promising strategy for unraveling pivotal mechanisms in MM progression, offering the potential for holistic and precise improvements in treatment strategies.

In this introduction, the authors discuss the multi-step process of multiple myeloma (MM) evolution from premalignant phases (MGUS and SMM) to symptomatic MM. They underscore the challenge of predicting progression from MGUS/SMM to MM, emphasizing the importance of intervening at the premalignant stages. The concept of cancer stem cells (CSCs) is explored in the context of MM, noting the debate over their existence. The authors delve into the multistep and multifaceted process of malignant transformation, driven by clonal evolution, genetic lesions, and interactions with the tumor microenvironment.

The authors review the exploration into tumor-initiating cells in MM, focusing on phenotypic characteristics. They discuss the speculation that myeloma-initiating/progenitor cells contribute to disease recurrence. The role of circulating CD19+ B cells with chromosomal abnormalities and resistance to chemotherapy is highlighted, suggesting their importance as a therapy-resistant tumor reservoir. The authors examine the controversy surrounding the self-renewal ability of CD138- B cells in MM and the significance of side population (SP) cells with stem-like phenotypes.

The phenotypic characteristics of myeloma-initiating cells are explored, including the potential relevance of CD24 as a marker. The authors point out the importance of advanced methodologies, such as high-throughput methods and single-cell sequencing, in investigating minor progenitor populations. They emphasize the need for a more comprehensive understanding of the phenotypic characteristics of myeloma-initiating cells to identify therapeutic targets and improve treatment outcomes.

In the final section, the authors highlight the advancements in single-cell sequencing technology, providing a higher resolution clonal map of MM cells and offering a comprehensive analysis of the mechanisms underlying malignant transformation. They suggest that these novel insights into myeloma initiation could lead to improved therapeutic strategies and ultimately contribute to the goal of curing the disease and preventing relapse.

The provided text is an extensive scientific review of multiple myeloma (MM), covering various aspects of its molecular and cellular biology. To facilitate a peer review, let's break down the text into key points, highlight limitations, and suggest improvements:

1.    Strengths:

    • Comprehensive Literature Review: The review extensively covers recent studies and research findings related to MM, incorporating various techniques such as scRNA-seq, scATAC-seq, WGS, and WES.
    • In-Depth Exploration: The text delves into different aspects of MM progression, including clonal evolution, drug resistance, alterations in the tumor microenvironment, and potential therapeutic targets.

2.    Limitations and Suggestions for Improvement:

·       Clarity and Structure:

    • Issue: The text lacks clear section headers, making it challenging for readers to navigate and identify specific topics.
    • Suggestion: Introduce clear section headers for each major topic (e.g., "Clonal Evolution," "Drug Resistance," "Tumor Microenvironment") to enhance readability.

·       Redundancy:

    • Issue: There's some repetition of information, particularly regarding the techniques used (e.g., scRNA-seq, WGS) and general statements about the importance of understanding MM.
    • Suggestion: Consolidate repetitive information and focus on presenting key findings in a concise manner.

·       Citation Format:

    • Issue: The text lacks a standardized citation format, making it challenging to trace specific references.
    • Suggestion: Adopt a consistent citation style (e.g., APA, MLA) to improve clarity and enable readers to locate the referenced studies easily.

·       In-Text Figures:

    • Issue: The text is dense, and the inclusion of in-text figures or diagrams could aid in visualizing complex concepts.
    • Suggestion: Consider incorporating relevant figures or diagrams to illustrate key processes, pathways, or findings.

·       Interconnection of Ideas:

    • Issue: The transition between different topics could be smoother, and the interconnection of ideas between sections needs improvement.
    • Suggestion: Use transitional sentences or paragraphs to better connect ideas and maintain a logical flow between sections.

·       Discussion of Methodological Limitations:

    • Issue: The review lacks a discussion of potential limitations associated with the methodologies used in the referenced studies.
    • Suggestion: Include a section discussing the limitations of the methods employed in the studies, addressing issues such as sample size, biases, or technological constraints.

·       Conciseness:

    • Issue: The text is quite extensive, and some sections may benefit from concise summarization.
    • Suggestion: Where applicable, provide concise summaries of key findings to maintain reader engagement and clarity.

Incorporating these suggestions should enhance the overall readability, cohesiveness, and impact of the scientific review 

The manuscript can be reconsidered after major revisions.

Author Response

Thank you for you review. Please see the attachment.

Reviewer 2 Report

Comments and Suggestions for Authors

In the manuscript “Novel insights into initiation and evolution of multiple myeloma by multi-omics investigation” Gong et al. aim at providing an overview of the most recent omics-based findings to elucidate myeloma initiation and progression. This is an extremely challenging and complex task. Indeed, despite including all relevant literature, the review is not well structured and sometimes confusing.  

To improve clarity, I strongly recommend following the PRISMA checklist for systematic review report.  Specifically, the manuscript should include: method section (search strategy), additional figures to summarize the ideas (f.e. changes during clonal evolution of MM), subdivisions within the paragraphs to improve the structure.

In addition, the paragraph “Phenotypic characteristics of tumor initiating cells in MM” should be removed since it does not include any omics study. Instead, I would recommend a short paragraph introducing the technical differences of omics studies: scRNA, sc-ATAC-RNA, proteomics, metabolomics etc.

Finally, the conclusion should  be optimized to discuss the challenges and future perspectives of omics in MM.

Minor comments:

-          What does Table 1 represent? Is it a summary of the most relevant studies using omics? Not all the studies summarized in table 1 have been discussed in the text. Why?

-          The sentence “Transcriptional adaptation preceded detectable outgrowth of genetically discernible drug-resistant clones and was associated with widespread enhancer remodeling”. (line 341/342) is copied from the abstract and has no meaning out of its context.

-          The sentences in line 74/75 page 3 and page 4, line 126/127 are the same. 

-          Rephrase the sentences in line 241/242, 330/331.

-          The following information are missing: figure legend, Title of the table

Comments on the Quality of English Language

I recommend editing of english language 

Round 2

Reviewer 2 Report

Comments and Suggestions for Authors

Thank you for reviewing the manuscript. I have no additional comments. 

Comments on the Quality of English Language

No further comments on the quality of English Language